# Material Characterization of Locally Available Textile Fabrics for Structural Applications

Safeer Abbas [1], Rizwan Amjad [2], Moncef L. Nehdi [3,*] and Shahid Ali [4]

1  Department of Civil Engineering, University of Engineering and Technology, Lahore 54890, Pakistan
2  Department of Civil Engineering, NFC Institute of Engineering and Fertilizer Research, Faisalabad 38090, Pakistan
3  Department of Civil Engineering, McMaster University, Hamilton, ON L8S 3L8, Canada
4  Department of Civil Engineering, National University of Computer and Emerging Sciences (FAST-NU), Lahore 54770, Pakistan
*  Correspondence: nehdim@mcmaster.ca

**Abstract:** In the current era, rehabilitation and strengthening of reinforced concrete structures is a major need due to premature structural damage owing to various environmental effects, natural hazards and major modifications in the existing building use. Textile fabrics can be an economical and viable option in comparison to traditional strengthening techniques. Therefore, this study was planned to investigate the use of locally available textile fabrics for structural applications leading to economical and sustainable solutions. Sixteen fabrics were collected randomly from the local market and a series of tests including microscopic analysis, mass per unit area, ends and picks count, yarn number and uniaxial tensile strength were conducted to explore the most suitable textile fabric from strength and application aspects. Moreover, rectangular textile-reinforced mortar specimens were prepared incorporating those textile fabrics. Tested textile fabric specimens exhibited mass per unit area in the range of 117 to 1145 g/m$^2$ depending on the fabric types. It was observed that tensile strength of the tested textile fabric depends on fiber composition, ends and picks count, yarn number and weave type. The greater the number of yarns in a fabric, the denser it will be and therefore it will be stronger in either direction (warp and weft). It was observed that the tensile strength in warp direction was higher than in weft direction due to the higher number of yarns in the warp direction. For instance, tested specimen TF16 showed ultimate tensile loads of 2890 and 2600 N in warp and weft directions, respectively. Furthermore, plain weave type fabric showed higher strength compared to that of the twill weave. It can also be argued that among the sixteen selected fabric specimens, plain weave fabric (i.e., glass) was found most suitable for textile-reinforced mortar applications due to adequate spacing and alternative movement of yarns, which leads to a stronger bond with the matrix and ultimately achieving higher tensile strength.

**Keywords:** reinforced concrete; textile fabrics; plain weave; twill weave; warp and weft directions; textile reinforced mortar

## 1. Introduction

Reinforced concrete (RC) is the major part in the urban infrastructure. Steel reinforcement resists the tensile stresses and concrete provides the compressive strength. In order to ensure the longevity of RC structures, their durability performance needs to be maintained with respect to corrosion of steel reinforcement and concrete deterioration. Due to deterioration of RC structures, retrofitting and strengthening is direly needed in the current era. Traditional techniques for the rehabilitation of RC include strengthening through steel strips and fiber reinforced polymer (FRP) sheets [1,2]. However, both these techniques have major drawbacks associated with their durability and uneconomical issues. Therefore, innovative techniques and materials are needed for more durable and economical rapid strengthening of RC structures.

Textile reinforcement is relatively a newer technique for strengthening and rehabilitation of RC structures. It may consist of unidirectional or bi-directional fibers along with various composite materials [3–7]. Furthermore, textile composite is made up of more than one yarn in a unique manner for the desired properties including high strength and heat resistance. There are four types of commonly used fabrics [8]: woven, knitted, braided and stitched. The difference between these types is dependent on the yarn movement and grip of yarns. Woven fabrics are made up of two sets of interlaced yarn at a 90-degree angle to each other by crossing over one another. Knitted fabric is manufactured by inter-looping yarns in a vertical or horizontal direction. Braided fabrics are made up of interlacing sets of continuous yarns and in stitched fabric, yarns are stitched together [8].

Various studies have been conducted in the past regarding the mechanical properties of specimens incorporating various textile fabrics. For instance, Portal et al. [9] investigated the tensile response of glass, basalt and carbon textile materials under accelerated exposures. It was reported that carbon textile fabrics had shown enhanced performance under aging exposure. Valeri et al. [10] studied the tensile behavior of textile-reinforced concrete (TRC) incorporating carbon textile fabrics. It was observed that crack spacing in the tested TRC specimens was dependent on the amount of reinforcement and roving distance [10]. Neves and Felicissimo [11] also studied the crack control behavior of TRC with various type of matrix incorporating unresin carbon fibers. Karnoub et al. [12] compared two types of fabrics with respect to weave type. Results showed that woven fabric had much more tensile strength as compared to the knitted fabric. Jahan [13] concluded from her study that the tensile strength of fabric depends on geometry, type of fiber and testing arrangement. The tensile strength of warp direction yarns is mostly higher than weft direction yarns. Due to lower porosity and more density of plain weave fabric, it had more strength than that of the twill fabric. Plain weave had the highest strength in warp direction due to more density of weft yarns in comparison to other weave types. Malik et al. [14] concluded that the weave structures play an important role in the tensile strength of woven fabric. From the results, it was concluded that polyester–cotton plain weave had more strength than that of 3/1 twill fabrics in both directions (warp and weft). Blanksvard et al. [15] conducted a study on the weave type, spacing between warp and weft, on the TRM strengthening system. Three different textile fabrics manufactured from carbon fibers were used. Results indicate the high strength TRM mechanism shifted the mode of failure from shear to flexure. Smaller grid spacing resulted in the first shear crack at higher load, and it also gave better results in re-arrangement of cracks. Escrig et al. [16] investigated on RC beams which were strengthened with different textile fabrics for examining the shear behavior. Glass, polybenzoxazole, basalt and carbon materials were used as textile fabrics. In that study, different types of mortar mixtures were used as a binding material. It was concluded that the bond between textile fabrics and concrete or mortar materials considerably influenced the structural performance of the overall system. Tzoura and Triantafillou [17] carried out a study on the TRM system with variation in the type of textile fabrics. The used textile fabrics were of two types, light and heavy in terms of weight, and also with variation in number of layers. Light and heavy fabrics had different numbers of fibers but were equal with regard to the number of yarns in warp and weft directions. Mass per unit area for lighter fabric was 174 g/m$^2$ while for heavy fabric it was 348 g/m$^2$. Results revealed that higher strength was achieved with heavy fabric (133% higher strength was achieved). In the case of the two-layered system, 183% higher strength was attained with heavy fabric. Al-Salloum et al. [18] conducted research on TRM composites with changing mortar types: cementitious mortar and polymer modified mortar. When two TRM layers were used, there was no major variation in strength. However, in the case of four layers of TRM, polymer modified mortar showed higher strength results (16%) compared to an identical specimen with two layers of TRM. Larbi et al. [19] conducted research on the variation of thickness of mortar (5 and 10 mm) in TRM and found no significant difference in the gain of strength for tested thicknesses. Triantafillou and Papanicolaou [20] conducted experimental work on TRM and varied the number of fabric layers. It was concluded that a single layer fabric

resulted in abrupt shear failure; however, TRM failure mode changed to flexure for a double layer of fabrics. Tetta et al. [21] tested beams in flexure with variation in layers of TRM from one to three. In the case of two layers of TRM, there was a 57% increase in strength as compared to one layer of TRM. Similarly, a 92% increase in strength was observed when three layers were used in comparison to an identical specimen with one layer of TRM.

Stolyarov and Ershov [22] conducted an experimental study on plain weave polyester multifilament yarns with 18.4 yarns/centimeter. Results showed greater strength in the warp direction while higher elongation in the weft direction. Chairman et al. [23] investigated how the geometry of fabric played a significant role in mechanical properties. This study was conducted on basalt plain and twill fabric. Zhou et al. [24] worked on the plain and twill weave fabrics and found an effect of the weave pattern on the mechanical performance. Composite with twill weave having lesser areal density resulted in higher elastic modulus and breaking strength than higher areal density weave composites. Zhang et al. [25] conducted a study on textile-reinforced concrete using high ductile basalt fibers and found that increased grid level improved the stress-strain behavior of composite concrete. Zhang et al. [26] conducted a study on textile-reinforced composites with highly ductile fiber along with carbon fibers in order to cope with the brittleness of the mortar matrix. The addition of carbon fibers in the mortar mixture reduced the crack width spacing [26]. Begum and Milasius [27] reported that the weave pattern has a vital role in the mechanical and physical properties. For instance, higher elongation was found in plain weave fabric compared to twill weave fabric due to higher interlacement. Erbil et al. [28] worked on three different weft yarns based on 100% cotton, core-spun, and dual-core-spun yarn. Ferrara et al. [29] conducted a study on the flax-based textile fabric in textile-reinforced lime mortar with variation of reinforcement amount and impregnation treatment. Impregnation treatment improved the tensile strength of textile-reinforced mortar. Moreover, increasing the reinforcement ratio increased the strength and reduced the crack width. Torres et al. [30] tested masonry walls reinforced with textile-reinforced mortar based on glass fabrics for retrofitting. Strengthened masonry walls achieved much higher compressive strength compared to unreinforced masonry walls. Moreover, the strengthened wall showed more ductility under cyclic loading. Tran et al. [31] developed an eco-friendly textile-reinforced concrete incorporating the industrial waste. It was reported that due to the incorporation of short textile fibers, significant improvement in mechanical properties and resistance against spalling was observed.

The potential of textile reinforcement has been well studied in previous research for its application in RC structures. However, previous research was mainly focused on the commercially available conventional textile reinforcement grid systems. Very scant literature is available on the unconventional textile fabrics available in the local market at economical rates. Therefore, this research work was mainly conducted on sixteen various types of textile fabrics procured from the local market. Various tests including microscopic analysis, mass per unit area, ends and picks, and tensile strength were conducted on textile fabrics to examine their properties for use in structural applications. Moreover, textile-reinforced mortar (TRM) strips were casted and tested to examine the tensile behavior of developed composite. This study made an effort to highlight the potential of unconventional textile fabrics for numerous structural applications and it will assist infrastructure stakeholders in using this economical and durable technique for the strengthening and rehabilitation of RC structures.

## 2. Materials and Mixture Proportions

Sixteen textile fabrics were selected from various sources in the local market for exploring their physical and mechanical properties. The selected fabrics are those most commonly used in local industry for a variety of applications. Moreover, to examine their viability for civil engineering purposes, plain and twill weave fabrics were selected. Initially, $1.5 \times 2.5$ m of textile fabrics from each source was procured. Ordinary Portland cement (OPC) was used for casting of textile-reinforced mortar (TRM) specimens. Fine sand and

marble powder were used as fine aggregates. The maximum size of marble powder used was 150 μm. Silica fumes, a supplementary material, was used in this study. It has very fine particles less than 1 μm. Table 1 shows the chemical properties of the materials used. Table 2 shows properties of the high-range water reducer used. Ordinary water was used for mixing purposes.

**Table 1.** Chemical properties of used cement, marble powder and silica fumes.

| Materials | $SiO_2$ | CaO | $Al_2O_3$ | $Fe_2O_3$ | MgO | $Na_2O$ | $K_2O$ |
|---|---|---|---|---|---|---|---|
| Cement (%) | 18.22 | 66.24 | 5.76 | 3.42 | 1.71 | 1.50 | 1.25 |
| Marble powder (%) | 1.10 | 56.22 | 0.97 | 0.10 | 0.51 | 0.06 | 0.11 |
| Silica fumes (%) | 84.12 | 1.32 | 0.46 | 0.61 | 0.79 | 0.48 | 1.32 |

**Table 2.** Properties of high-range water reducer.

| Properties | Values |
|---|---|
| pH | 6.8 |
| Viscosity | 125 cps |
| Density | 1.15 kg/L at 25 °C |
| Type | Carboxylic acid derivatives |
| Form | Whitish pale liquid |

Table 3 shows the mixture proportion used for casting the TRM specimens incorporating textile fabrics. Mortar mixing was done using an electric mortar bowl mixer at the rate of 50 rpm. The mixing bowl was first cleaned and moistened with water before adding materials into it. The mixing sequence was as follows: marble powder was added in the mixture along with sand and silica fumes and mixed for 30 s. Afterwards, cement was added and mixing continued for 1 min. Water was mixed with superplasticizer and then slowly added in the mixture while mixing continued. After achieving homogenous mixture, the flow test was performed and mortar paste was placed in molds and casted for the required number of TRM specimens. Three specimens of a mortar cube (50 mm × 50 mm × 50 mm) were casted and cured in water. After 28 days, specimens were taken out from the water curing and tested for compressive strength. Average mortar compressive strength was 38 MPa at 28 days. Figure 1 shows pictures of the selected sixteen textile fabrics.

**Table 3.** Mortar mixture proportion for casting of TRM specimens.

| Materials | Cement | Sand | Marble Powder | Silica Fume | Water | Superplasticizer |
|---|---|---|---|---|---|---|
| Quantities/cement mass | 1.00 | 0.50 | 0.65 | 0.14 | 0.28 | 0.11 |

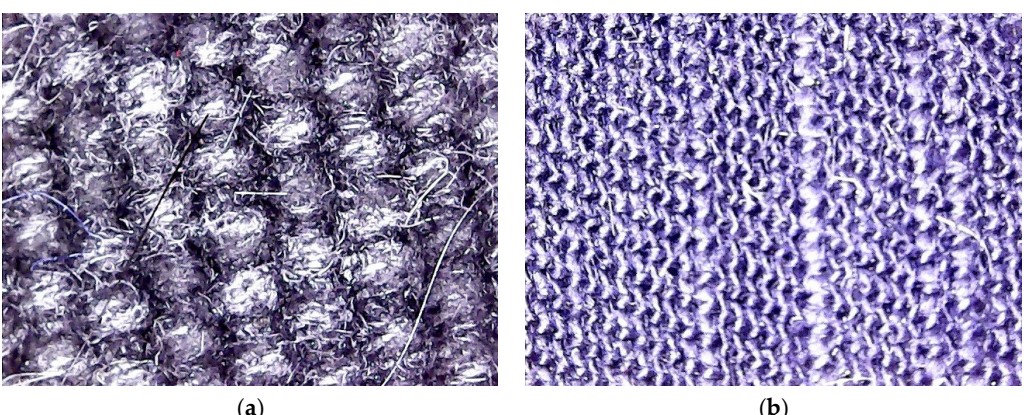

(a)                                    (b)

**Figure 1.** *Cont.*

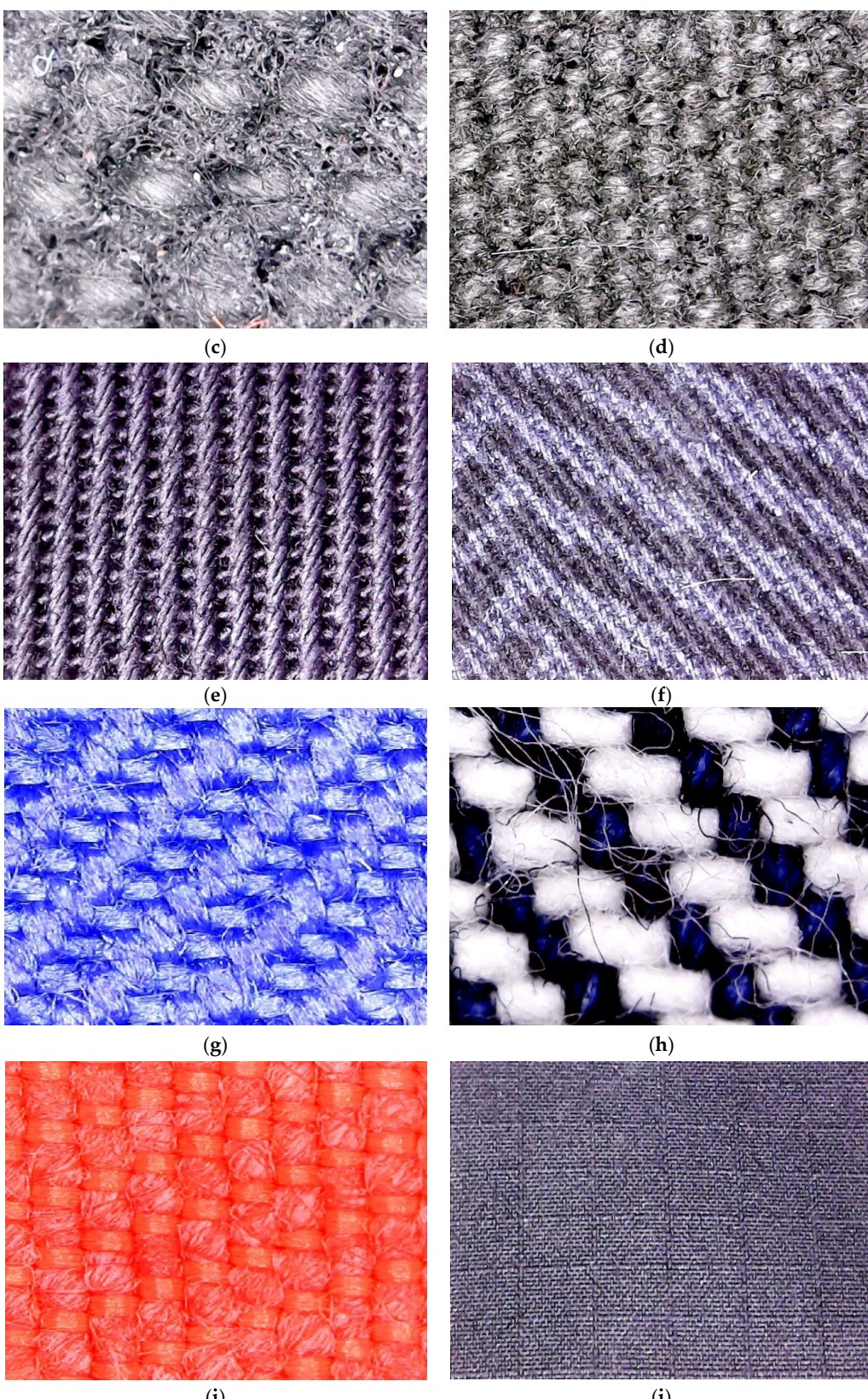

**Figure 1.** *Cont.*

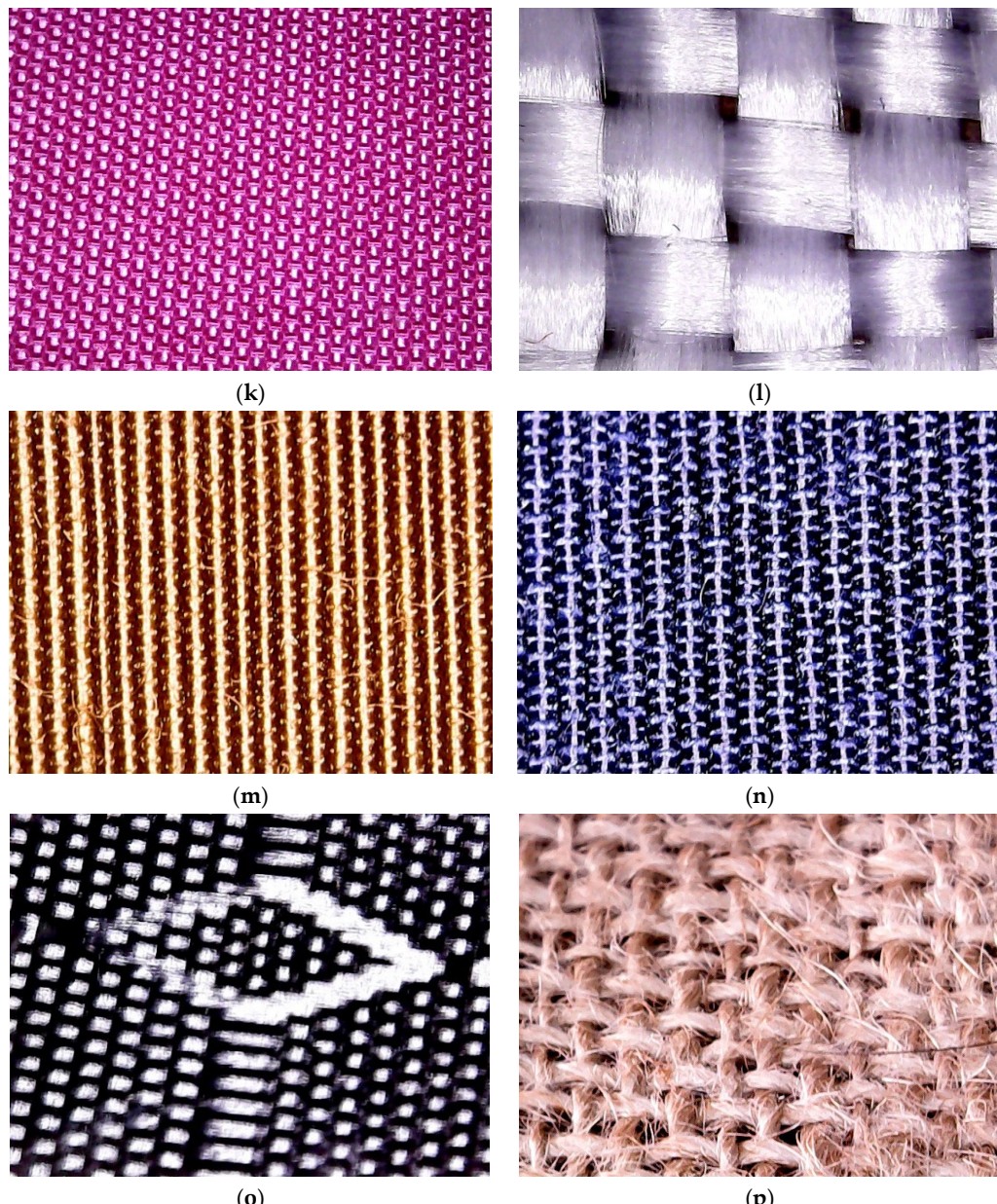

**Figure 1.** Pictures of selected textile fabrics (horizontal direction shows the warp direction and vertical direction represents the weft direction (**a**) TF1 (**b**) TF2 (**c**) TF3 (**d**) TF4 (**e**) TF5 (**f**) TF6 (**g**) TF7 (**h**) TF8 (**i**) TF9 (**j**) TF10 (**k**) TF11 (**l**) TF12 (**m**) TF13 (**n**) TF14 (**o**) TF15 (**p**) TF16.

## 3. Experimental Methodologies

Fabric specimens were placed in an environmental chamber at 20 °C and 65% relative humidity for 8 h before performing any test. First of all, microscopic analysis was performed on textile fabrics to evaluate the micro-structure and individual fiber orientation of used fabrics in warp and weft directions. Mass per unit area of all fabric specimens was determined in accordance with ASTM D3776 [32]. After conditioning, five specimens were cut from each fabric. Specimens of size 152 cm$^2$ were cut with a fabric sample cutter. The specimen weights were taken and calculated for the mass per unit area (g/m$^2$). Figure 2 shows the fabric cutter and cut specimens for mass per unit area. Ends (warp) and picks (weft) were determined according to ASTM D3775 [33]. Ends and picks were counted over 1-inch space and results were presented as warp yarns (yarns/inch) × weft yarns (yarns/inch). This test was performed with pick counter graduated in inches up to 1/8 inch (Figure 3). Firstly, the warp side of specimen was raveled until full length yarns

had appeared. The pick counter was placed at five random places along the length of fabric and observations were taken. Thereafter, the average of the five observations was calculated. Counting of yarns was done in both (warp and weft) directions and five observations were taken in each direction. Table 4 shows the number of specimens or observations for each test performed.

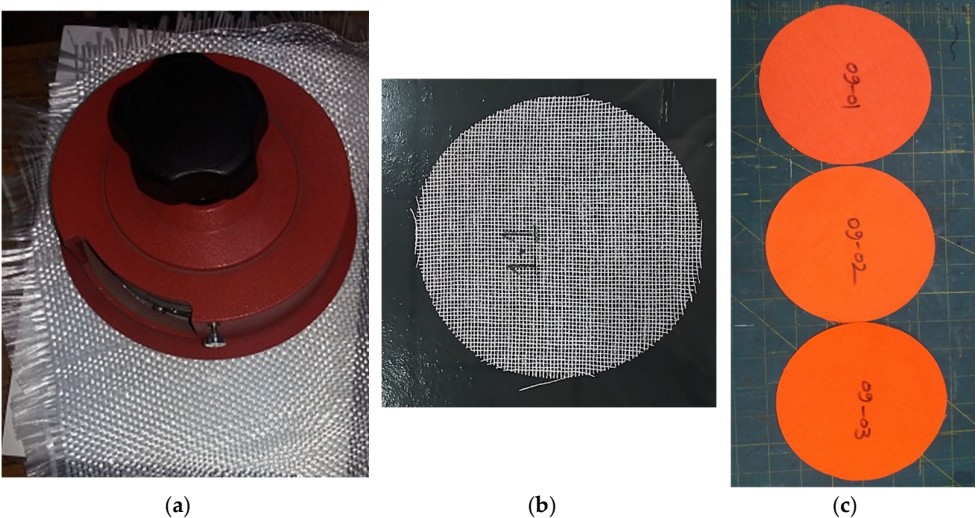

|  (a) | (b) | (c) |

**Figure 2.** Fabric specimens for mass per unit area (**a**) fabric cutter (**b**) TF12 specimen (**c**) TF9 specimens for mass per unit area.

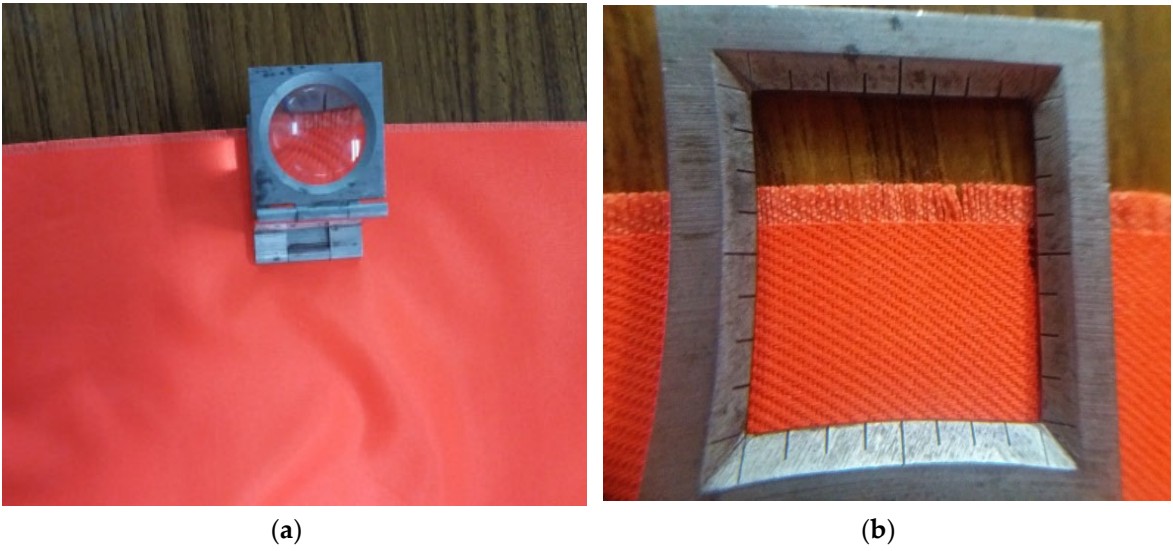

|  (a) | (b) |

**Figure 3.** Counting of ends and picks (horizontal direction is the warp direction) (**a**) pick counter (TF9 specimen) (**b**) graduations of pick counter with raveled fabric (TF9 specimen).

**Table 4.** Test matrix.

| Tests | Number of Specimens/Observations |
| --- | --- |
| Tensile test on textile fabrics | 3 |
| Ends and picks count | 5 |
| Mass per unit area | 5 |
| Fiber composition | 1 |
| Yarn count | 5 |
| Tensile test on textile-reinforced mortar strips | 3 |

Yarn number of all the tested textile fabric specimens was determined in accordance with ASTM D1059 [34]. Small patches of fabric of size 250 × 250 mm were cut from each fabric (Figure 4). Then, a significant number of full-length yarns were obtained from this small fabric patch in warp and weft directions. The yarns were trimmed to make them of equal length. Yarn weights were obtained by weighing on a balance and weight per unit length was calculated.

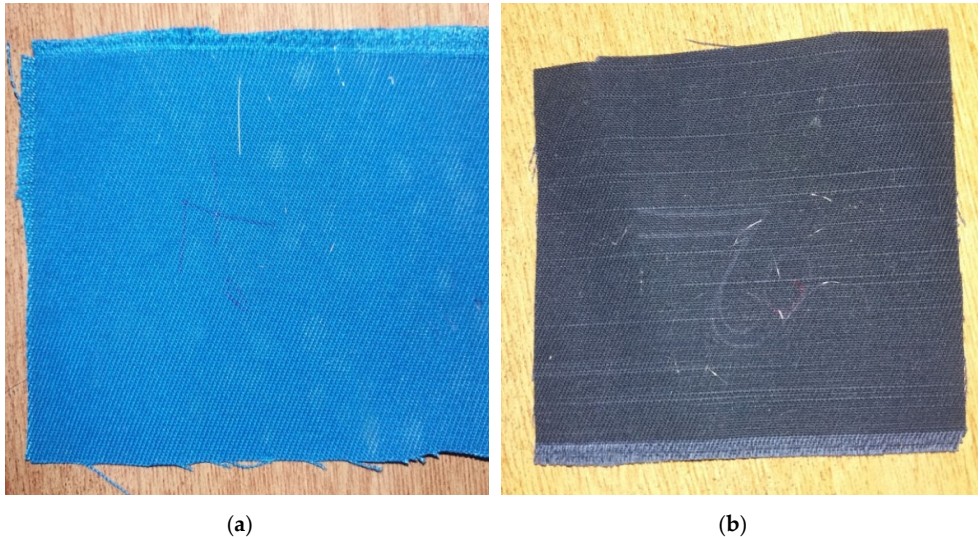

| (a) | (b) |

**Figure 4.** Specimens for yarn count test (horizontal direction is the warp direction), (**a**) TF7 specimen (**b**) TF2 specimen.

Tensile strength of fabric specimens was determined according to ASTM D5035 [35]. Three specimens having dimensions 500 × 50 mm were prepared in each direction (i.e., warp and weft) from each representative fabric specimen collected. This test was performed on a universal testing machine (UTM) with a capacity of 1000 kN (Figure 5).

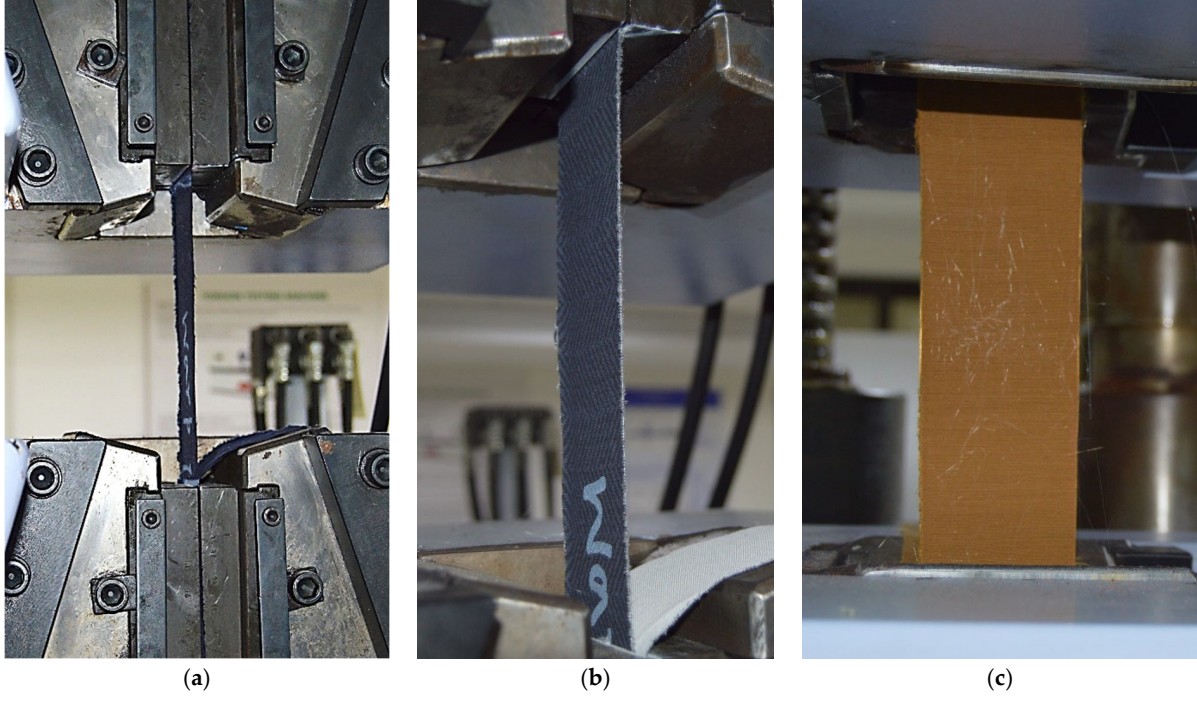

| (a) | (b) | (c) |

**Figure 5.** Testing setup for tensile testing of textile fabrics (**a**) TF6 (**b**) TF6 (**c**) TF13.

The dimensions of the upper and lower jaws of the UTM grip were 75 × 75 mm. The specimen was first placed in the upper jaws, with folding of the specimen on the other side of jaws, and a steel bar of 10 mm diameter was inserted in the folding to avoid slippage of specimens within the jaws. The same procedure was adopted with lower jaws as well. After placement of the specimen, gauge length was checked and we initiated the test. Observations of applied load and deflection were noted. Load was continuously applied until rupture of the test specimen.

Textile-reinforced mortar (TRM) strips of size 600 × 60 × 10 mm incorporating various types of textile fabrics were prepared. Sixteen types of TRM specimens were prepared for each type of textile fabric sample. Three specimens were casted for each fabric type. According to mixture design (Table 3), the mortar mixture was prepared. Oiling of the molds was done in which the TRM was to be casted. The first layer (around 5 mm) of mortar was poured with the help of a spatula (Figure 6a). After leveling the mortar, textile fabric was placed over that layer of mortar in the mold (Figure 6b). The placement of the fabric strip over mortar was carefully handled to prevent the disturbance of mortar thickness. After placing and leveling, a second layer of mortar was poured (Figure 6c). After two days, TRM specimens were taken out from their respective molds (Figure 6d) and placed in water for curing until testing age. A similar casting procedure for TRM was also reported in a previous study [36].

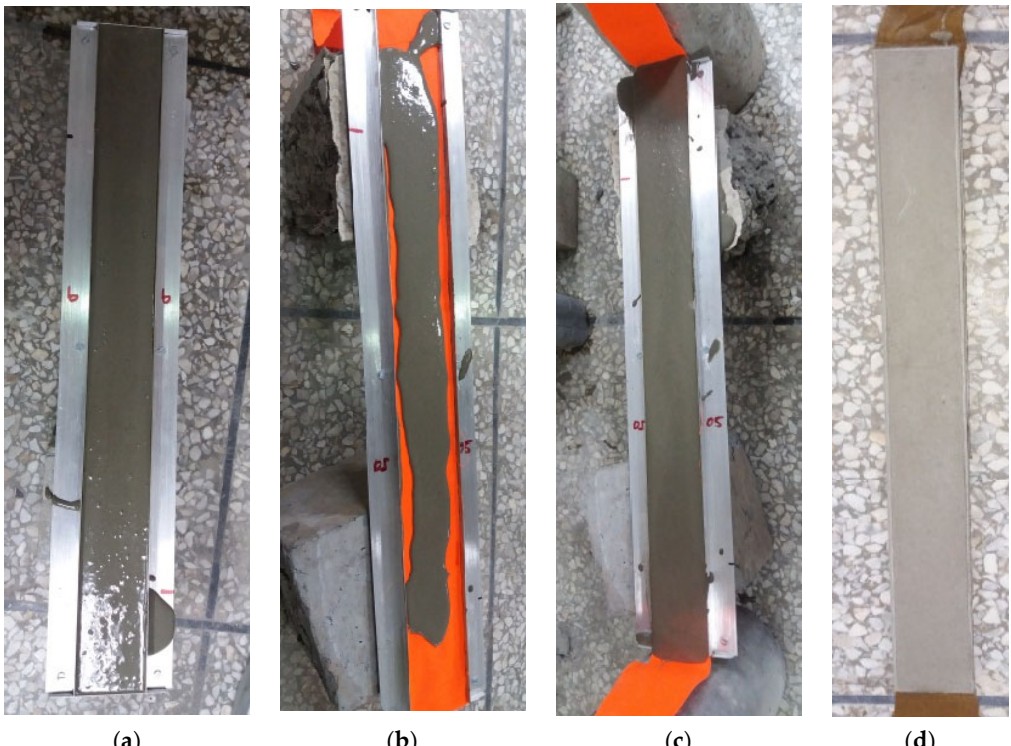

| (**a**) | (**b**) | (**c**) | (**d**) |

**Figure 6.** Casting of textile-reinforced mortar specimens (**a**) first layer of mortar, (**b**) placement of fabric strip, (**c**) second layer of mortar, (**d**) de-molded specimen.

Before testing, the ends of the casted TRM specimen were prepared (Figure 7a). Rubber pads of size 100 × 70 × 4 mm were prepared and glued over the ends of the specimen on both sides. These rubber pads prevented local failure within the grips and applied the pressure uniformly over the area. The specimen was first placed in the upper jaws of the UTM and then in the lower jaws. After placement of the specimen into the jaws, the test was started and observations were noted for loads and deflections. Figure 7b shows the placement of the TRM specimen in the UTM for testing purposes.

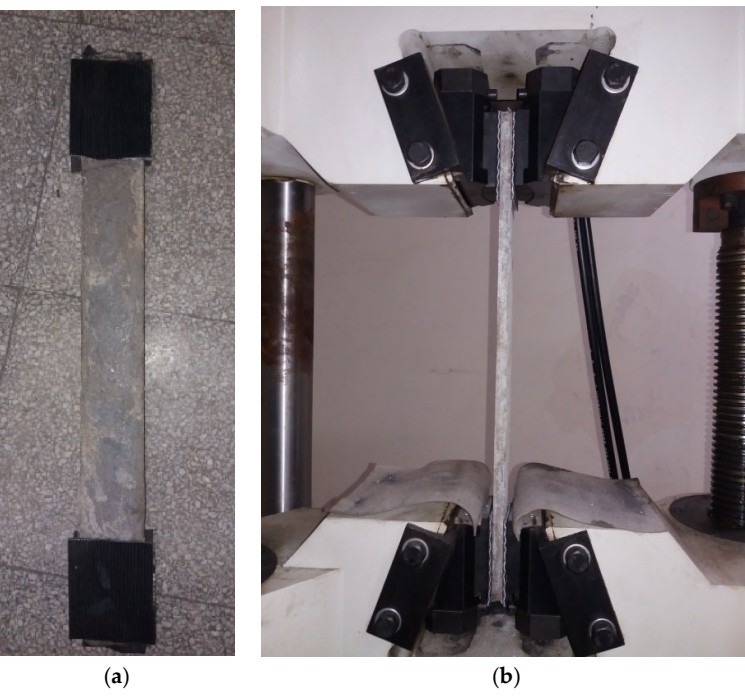

|       |       |
| :---: | :---: |
| (**a**) | (**b**) |

**Figure 7.** Testing setup for tensile testing of TRM specimens (**a**) TRM specimen with prepared ends (**b**) TRM specimen gripped in machine.

## 4. Results and Discussion

### 4.1. Textile Fabric Types

Every yarn in the weave is moving alternatively up and down to its perpendicular yarn. Three types of the most common weaves are used in the local fabric market: plain, twill (2/1) and twill (3/1) weave fabric. Plain fabric is the simplest type in which only one yarn alternatively moves up and down to its perpendicular yarn (Figure 8a). If weft yarn moves up and down to its perpendicular warp yarns with float of two, it is known as twill (2/1) weave fabric. Also, there is a shift in subsequent rows to create a diagonal pattern (Figure 8b). Similarly, for twill (3/1) weave fabric, weft yarn moves up and down to its perpendicular warp yarns with float of 3 (Figure 8c). This means that yarns are weaved with float of 3 and also a diagonal pattern. Table 5 shows the microscopic analysis of tested fabrics and examples of plain, twill (2/1) and twill (3/1) weave fabrics. Different magnifications (30 to 50×) have been employed for various tested fabrics to examine their weave type. Table 6 shows the fabric types based on quantitative analysis of fibers. Out of sixteen tested fabrics, five were plain, nine were twill (2/1) and two were twill (3/1) weave fabrics based on microscopic analysis, yarn distribution and orientation.

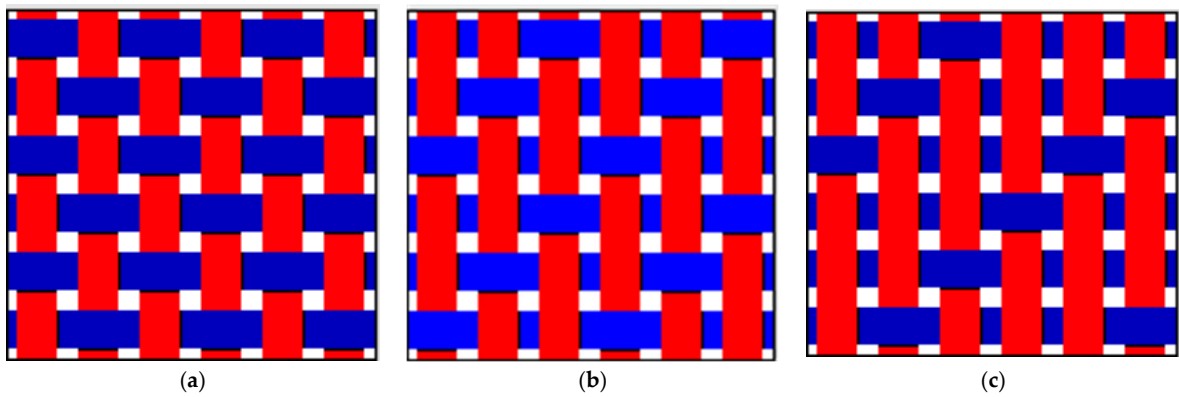

|       |       |       |
| :---: | :---: | :---: |
| (**a**) | (**b**) | (**c**) |

**Figure 8.** Schematic diagram of textile fabric types (red color indicates the warp direction) (**a**) plain weave fabric, (**b**) twill (2/1) weave fabric, (**c**) twill (3/1) weave fabric.

**Table 5.** Microscopic analysis of tested textile fabrics (vertical direction is the warp direction).

| Fabric Types | Microscopic Images | |
| --- | --- | --- |
| Plain weave fabric | 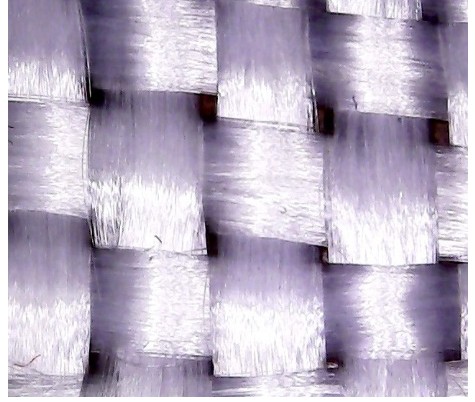<br>Glass fabric | 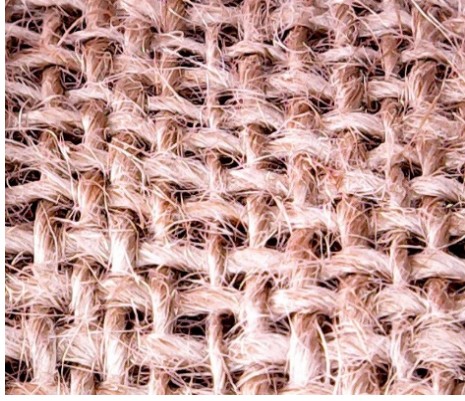<br>Jute fabric |
| Twill (2/1) weave fabric | 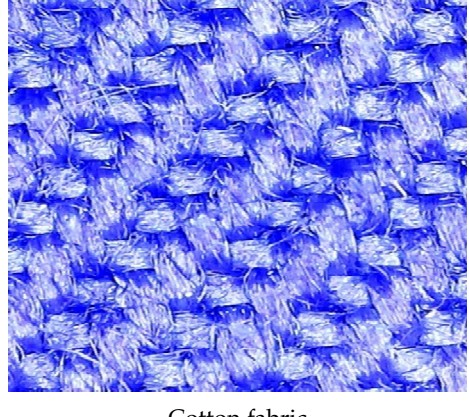<br>Cotton fabric | 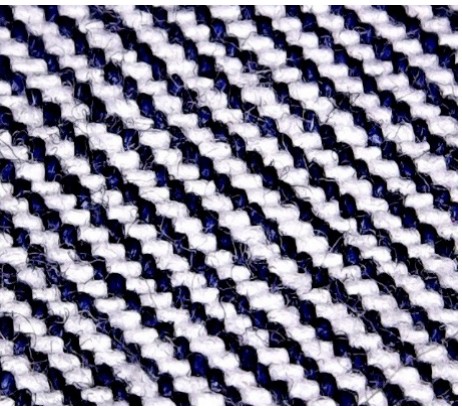<br>Denim fabric |
| Twill (3/1) weave fabric | 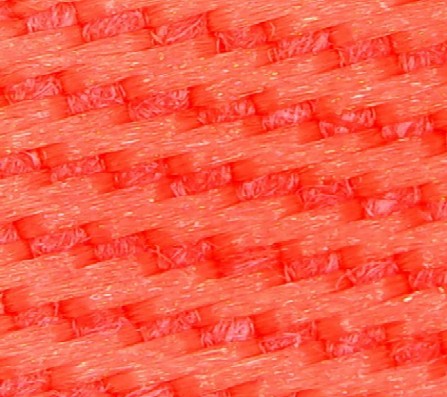<br>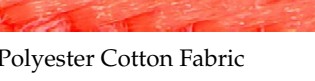Polyester Cotton Fabric | 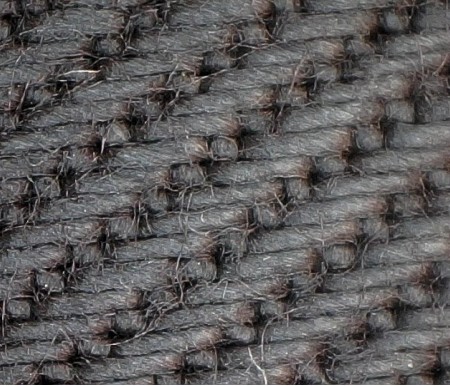<br>Cotton fabric |

### 4.2. Mass per Unit Area

Figure 9 shows the mass per unit area of tested fabric specimens. Results reported in Figure 9 were the average of five specimens with coefficient of variance (COV) less than 1%. Results showed that mass per unit was ranged from 117 to 1145 $g/m^2$ for the tested sixteen fabrics. This variation was mainly due to their various composition, fabric type, yarn thickness and numbers. TF1 has the highest mass per unit area (1145 $g/m^2$) and TF13 specimens had the lowest mass per unit area (117 $g/m^2$). This may be attributed to thicker yarns of specimen TF1 compared to TF13 specimens which had thinner yarns. Interestingly,

it was also noted that there was a higher number of yarns in the case of specimen TF13 in warp and weft directions, but still a lower mass per unit area. This indicated that mass was not only dependent on the yarn numbers, but also on the yarn thickness.

**Table 6.** Fabric types of tested textile specimen.

| Sample No | Weave | Composition | Yarn Count (tex) | Thickness of Fabric (mm) |
|---|---|---|---|---|
| TF1 | Twill 2/1 | Cotton | 507.15 | 1.70 |
| TF2 | Twill 2/1 | Cotton | 29.22 | 0.65 |
| TF3 | Twill 2/1 | Cotton | 318.48 | 1.43 |
| TF4 | Twill 2/1 | Cotton | 289.80 | 1.22 |
| TF5 | Twill 3/1 | Cotton | 50.32 | 0.85 |
| TF6 | Twill 2/1 | Cotton | 35.12 | 0.73 |
| TF7 | Twill 2/1 | Cotton | 48.42 | 0.83 |
| TF8 | Twill 2/1 | Cotton | 93.103 | 0.96 |
| TF9 | Twill 3/1 | Cotton and Polyester | 18.73 | 0.52 |
| TF10 | Plain | Polyester | — | 0.70 |
| TF11 | Twill 2/1 | Nylon | 56.77 | 0.87 |
| TF12 | Plain | Glass Fiber | 462.81 | 1.56 |
| TF13 | Plain | Polyester | 58.21 | 0.88 |
| TF14 | Plain | Nylon and Cotton | 62.24 | 0.90 |
| TF15 | Twill 2/1 | Polyester | — | 0.78 |
| TF16 | Plain | Jute | 290.71 | 2.67 |

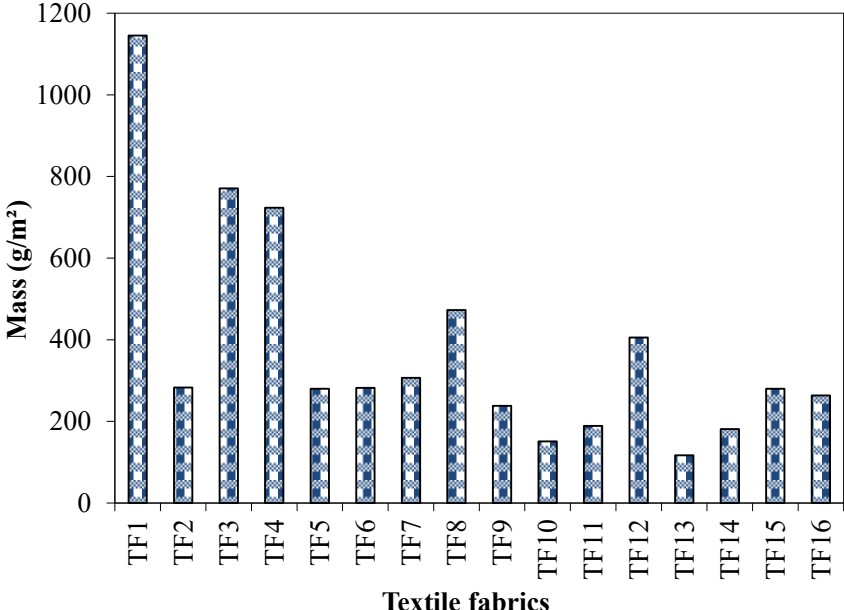

**Figure 9.** Mass per unit area results of tested textile fabrics.

Ferdous et al. [37] reported that the tested specimen of plain weave had 162 g/m$^2$ fabric weight. Similarly, El-Messiry et al. [38] found that cotton fabric, high tenacity polyester and polypropylene fabrics had fabric weights of 133 g/m$^2$, 130 g/m$^2$ and 87 g/m$^2$, respectively. High tenacity polyester had higher thickness compared to other tested specimens [28].

### 4.3. Ends (Warp) and Picks (Weft) of Fabric Specimens

Figure 10 shows the results of yarn density of tested fabrics in both warp and weft directions. The greater the number of yarns, the denser the fabric will be. For instance, specimen TF13 had 204 yarns/inch in warp direction and 80 yarns/inch in weft direction. This will lead to the highest yarn density in the tested fabrics. Tested fabric TF12 had the lowest yarn density. Yarn density plays a major role in the tensile strength of fabric.

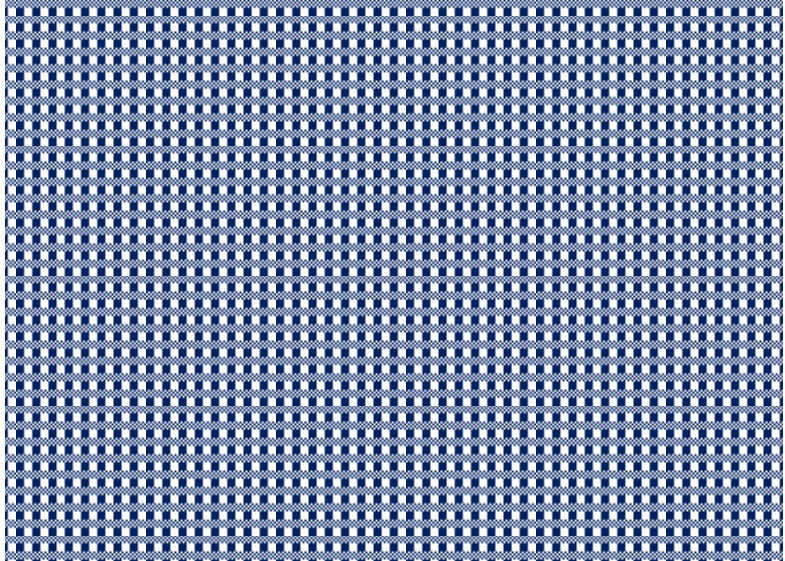

**Figure 10.** Results of ends (warp) and picks (weft) of tested fabric specimens.

Ferdous et al. [37] reported that, irrespective of the weave type for polyester-cotton specimens, their ends and picks densities were 77 and 43, respectively. Banerjee et al. [39] reported that cotton specimens had 100 and 60 ends and picks, respectively, in plain and twill weave.

### 4.4. Yarn Number

Figure 11 shows the results for yarn number based on shorter length specimens. Yarn number has a direct relation with yarn thickness. Specimen TF1 showed around 507 tex, which was the highest value for yarn number amongst all tested specimens. Specimen TF9 had the minimum value of yarn number (i.e., 18.73 tex). Specimen TF10 had very fine yarn which was not be separated and, therefore, its yarn count was not reported in Figure 11. Similarly, specimen number TF15 had coating over it, so yarn counting was not calculated for it. Yarn number played an important role in the tensile strength of fabric; the higher the yarn thickness, the higher the density will be. El-Messiry et al. [38] reported that yarn number for polypropylene fabric was 87 tex. Ferdous et al. [37] reported that yarn number for polyester–cotton fabric was 29.5 tex.

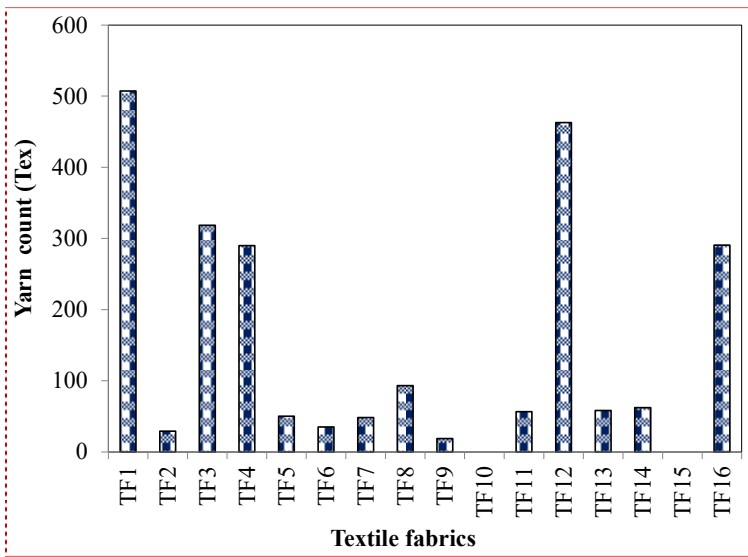

**Figure 11.** Yarn count of tested fabric specimens.

### 4.5. Tensile Behavior of Textile Fabrics

Figures 12 and 13 show the load-deflection curves of sixteen specimens for the tensile strength of fabric specimens in warp and weft directions. From these curves, it was observed that all specimens were attaining higher load at a very small deflection at the start of the test. Results have shown that plain weave had higher strength, followed by twill (2/1) weave fabric and twill (3/1) weave.

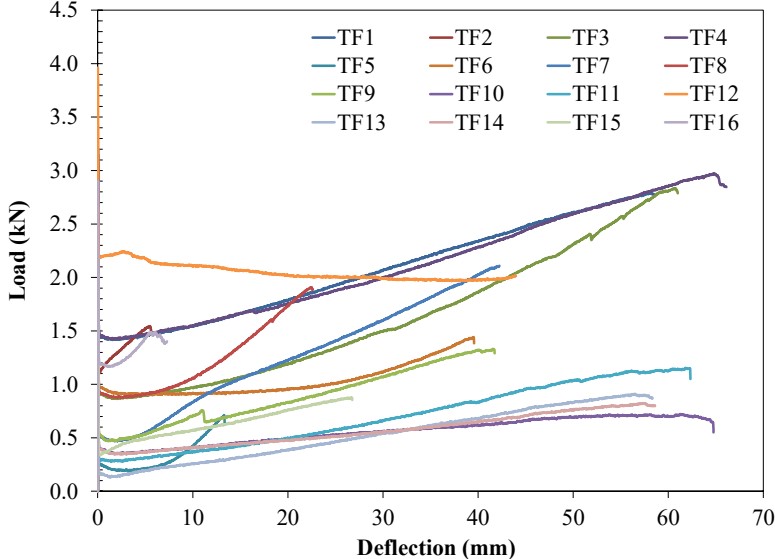

**Figure 12.** Load-deflection behavior of textile fabrics in warp direction.

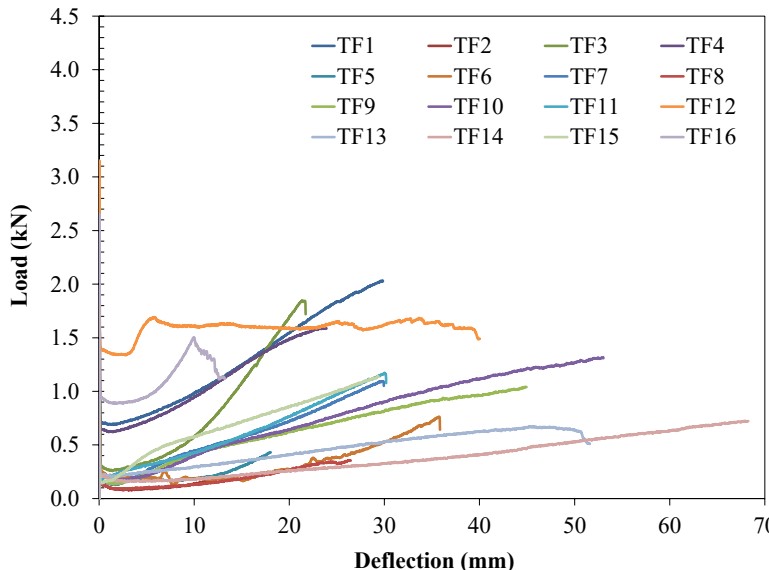

**Figure 13.** Load-deflection behavior of textile fabrics in weft direction.

Ferdous et al. [37] compared polyester–cotton twill and plain weave. Plain weave has maximum interlacement of yarns, which causes less slippage of fibers within yarns, making this the main reason for their lower strength. Twill weave has less interlacement so higher yarn float and stress distributes over more area, which gave more strength [37]. Banerjee et al. [39] reported that cotton twill weave had higher breaking strength than plain weave. The maximum number of interlacement of yarns and elasticity was in the plain weave; therefore, higher elongation occurred in plain weave. El Messiry et al. [38] reported that minor slippage of fibers within yarns occurred, which expressed in a lower breaking force in plain weaves than twill weaves. More interlacement of yarns caused

more concentration and riveting junctions of stress which causes a decrease in the strength of fabric.

Figures 14–16 show the results of the tensile strength of textile fabric specimens in warp and weft directions. These curves (Figures 14–16) were drawn up to 1 mm deflection to see the initial variation in load and corresponding deflection. The maximum deflection values before complete failure were very high. From these curves, it was observed that all specimens were taking high loads with a very small deflection. After the development of a first crack, an abrupt decrease in loading was observed. Thereafter, deflection was increasing with approximately constant load. The initial stiffness of tested similar fabric types (plain, twill (2/1) and twill (3/1)) was comparable. Table 7 shows the average tensile results of peak load and maximum deflection in warp and weft directions for tested fabric specimens. It was observed that the tested fabric has shown more strength in the warp direction compared to that of the weft direction (Table 7). This higher strength in warp direction was attributed to their higher number yarns in the warp direction than the weft direction. Moreover, due to the lesser number for the weft, interlacement of yarns will be less and less energy will dissipate through the warp yarns; hence, the strength will be higher in that warp direction. The TF12 specimen showed the maximum load in either direction due to higher yarn count, which is linked to the thickness of yarn. Furthermore, TF2, TF3, TF4 and TF6 showed comparable results in warp direction irrespective of yarn counts along with ends and picks.

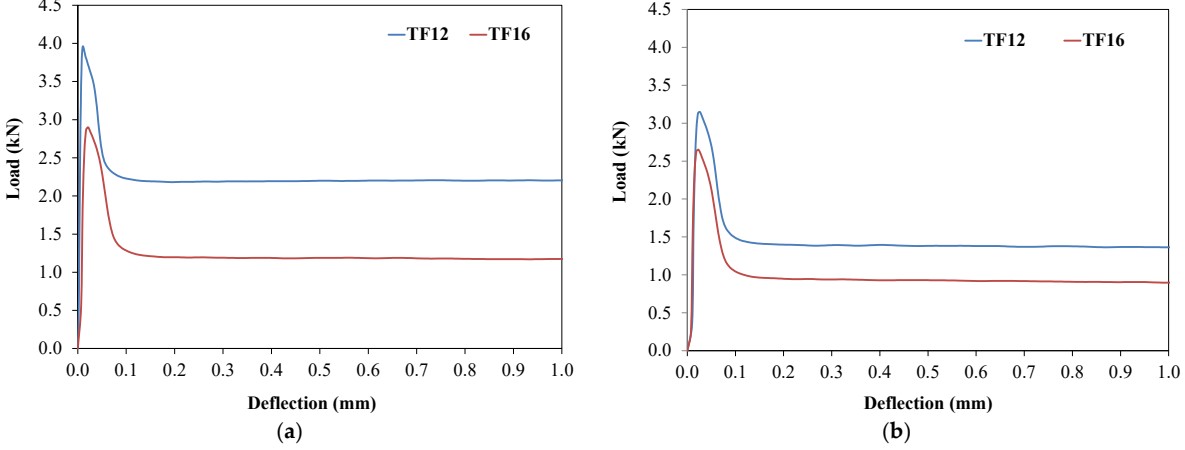

**Figure 14.** Load-deflection behavior of plain weave fabric (**a**) warp (**b**) weft.

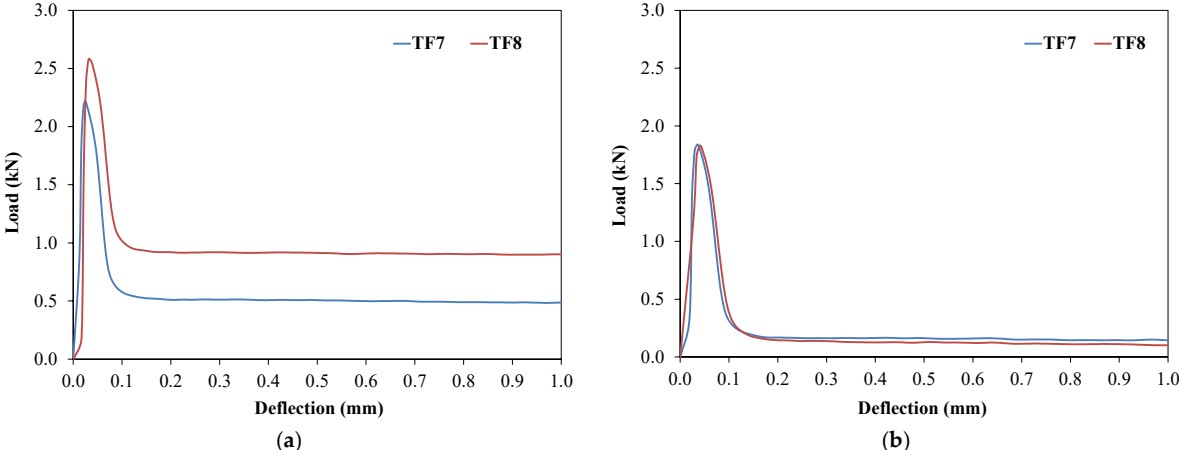

**Figure 15.** Load-deflection behavior of twill (2/1) weave fabric (**a**) warp (**b**) Weft.

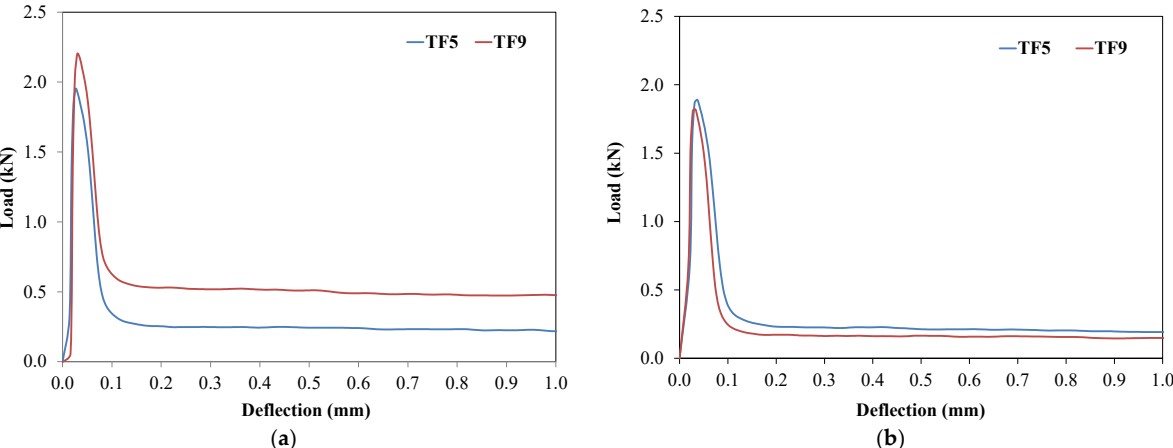

**Figure 16.** Load-deflection behavior of twill (3/1) weave fabric (**a**) warp (**b**) weft.

**Table 7.** Tensile results of tested textile fabrics.

| Specimens | Peak Load (N) | | Maximum Deflection (mm) | |
|---|---|---|---|---|
| | **Warp** | **Weft** | **Warp** | **Weft** |
| TF1 | 3178 | 2401 | 78.53 | 37.78 |
| TF2 | 2582 | 1456 | 12.97 | 23.50 |
| TF3 | 2531 | 1999 | 95.47 | 38.30 |
| TF4 | 2670 | 2240 | 62.61 | 33.93 |
| TF5 | 1915 | 1710 | 22.18 | 30.50 |
| TF6 | 2530 | 1945 | 33.28 | 30.43 |
| TF7 | 1960 | 1585 | 43.59 | 31.99 |
| TF8 | 2180 | 1730 | 28.27 | 37.56 |
| TF9 | 2185 | 1525 | 45.13 | 44.26 |
| TF10 | 2015 | 1680 | 70.69 | 60.57 |
| TF11 | 1995 | 1795 | 44.05 | 38.81 |
| TF12 | 3880 | 3125 | 43.96 | 43.87 |
| TF13 | 1820 | 1675 | 41.35 | 44.22 |
| TF14 | 2030 | 1765 | 44.22 | 44.34 |
| TF15 | 1955 | 1680 | 29.94 | 47.38 |
| TF16 | 2890 | 2600 | 44.20 | 39.58 |

Two specimens of the same weave were compared. In Figure 14, plain weave specimen TF12 showed higher strength in terms of load than specimen TF16 in warp direction. The reason might be that the specimen TF12 had higher thicker yarns than specimen TF16. Furthermore, specimen TF12 was manufactured from glass fiber, which is stronger, while specimen TF16 was produced from jute fibers. Ferdous et al. [37] reported that polyester–cotton plain weave showed 0.48 kN breaking force in warp direction. Banerjee et al. [39] reported that cotton plain weave had 0.52 kN breaking force in warp direction.

Similarly, twill (2/1) fabric type specimens were compared (Figure 15). Twill (2/1) specimen TF8 showed higher strength than specimen TF7 in warp direction. Maximum deflection recorded for TF8 and TF7 specimens was 28.27 mm and 43.59 mm, respectively, in warp direction. Specimen TF8 had more thicker yarns than specimen TF7. Ferdous et al. [37] reported that polyester–cotton twill weave had 0.57 kN breaking force in warp direction. El Messiry et al. [38] reported that longer floats with a lower number of intersections spread load over more area and this resulted in higher tensile strength. Figure 16 shows the results for twill (3/1) weave fabric specimens. Specimen TF9 resulted in higher strength than specimen TF5 in warp direction. Banerjee et al. [39] reported that cotton twill (3/1) fabric showed 0.59 kN breaking force in warp direction. El-Messiry et al. [38] concluded that the fewer the number of crossover points, the fewer will be the stress concentration points; hence, the higher the breaking strength will be.

Figure 17 shows the failure and rupture of tested textile fabric specimens. Plain weave fabric type achieved higher loads in comparison with twill (2/1) and twill (3/1) fabric types.

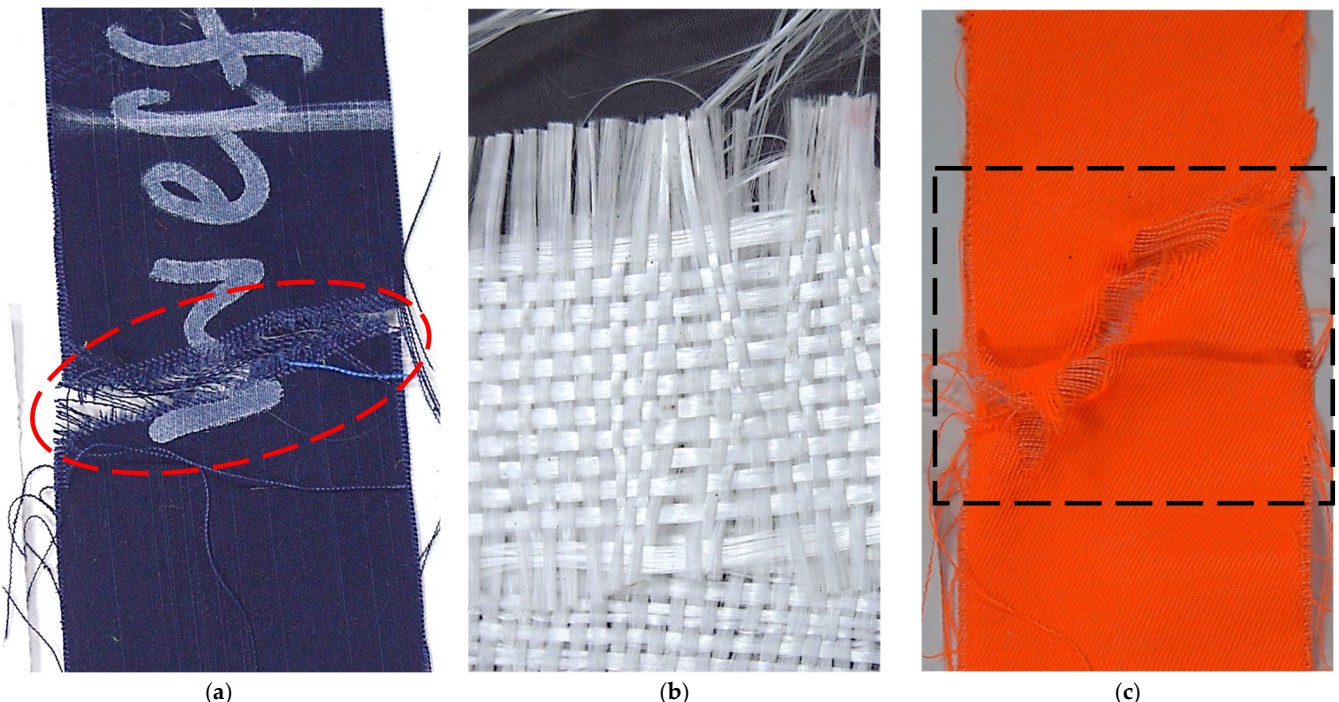

| (a) | (b) | (c) |

**Figure 17.** Failure of tested textile fabrics (**a**) TF2 (**b**) TF12 (**c**) TF9.

The tested plain fabrics have higher thicker yarns than other specimens. Tested twill (3/1) weave fabric had three times thinner yarns than twill (2/1); therefore, the strength of tested twill (2/1) was higher than the twill (3/1) specimen. Twill (3/1) had bigger float of interlacement than twill (2/1), which means less energy will be utilized to pull warp yarns, because interlacement of perpendicular yarns was distant. Plain weave has maximum interlacement of yarns, which causes less slippage of fibers within yarns, making this the main reason for the higher strength.

### 4.6. Tensile Response of Textile-Reinforced Mortar Strips

Figure 18 shows the tensile results of textile-reinforced mortar (TRM) specimens. Specimen TRM12 had the highest tensile breaking force of 4.18 kN. Specimen TRM10 had 0.47 kN breaking force which is the minimum amongst all the tested specimens. Specimen TRM12 had approximately nine times higher strength than specimen TRM10. Tested specimen TRM12 had thicker yarns and it was made up of glass fibers which exhibited higher strength. Specimen TRM10 had coating over it and also it was made of cotton fabric. Colombo et al. [36] stated that weft spacing played two important roles in the composite, i.e., when weft spacing was lower, it helped in warp delamination and crack propagation in weft direction, while at higher weft spacing, it prevented sliding of warp yarns. Also, lower spacing of weft caused premature cracks at very low strength due to reduced effective tension induced along warp direction.

Figure 19 shows the cracking pattern of a tested textile-reinforced specimen. The specimen shows delamination behavior. This behavior may be due to lesser spacing between fabric yarns and surface of fabric which caused delamination between fabric material and mortar mixture. The surface of fabrics plays a vital role in bonding. Rougher surface and yarn spacing lead to an improved bond behavior between fabric material and mortar mixture, leading to an increase in the tensile strength.

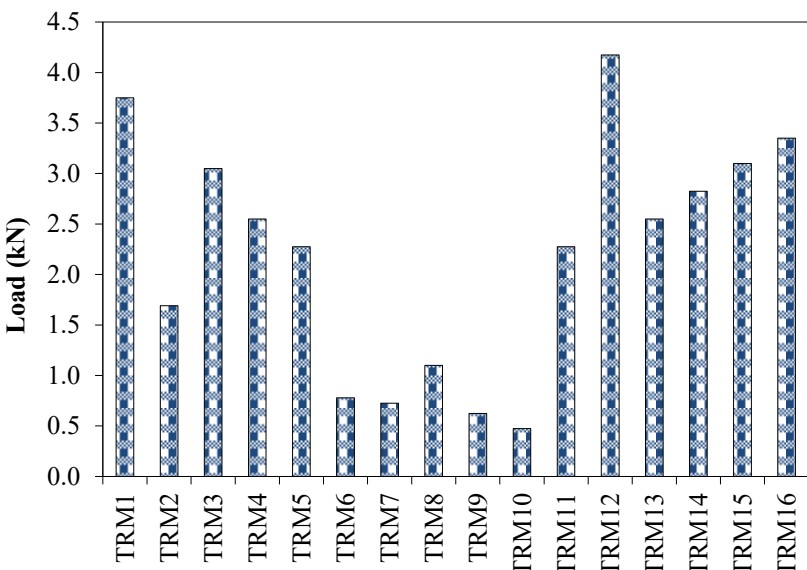

**Figure 18.** Tensile load carried by TRM strips incorporating various textile fabrics.

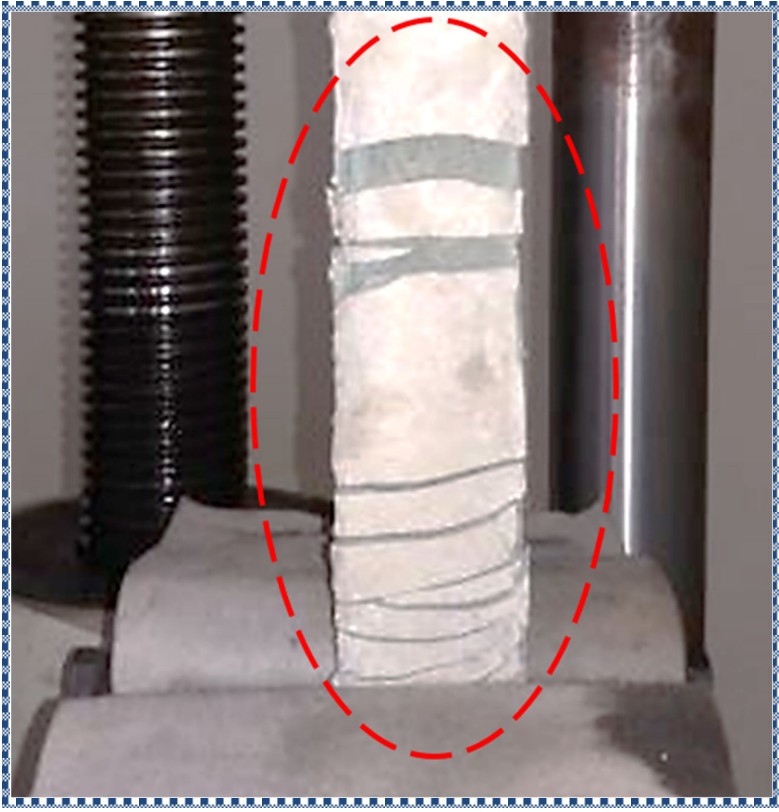

**Figure 19.** Crack pattern in tested TRM specimen (TRM2).

## 5. Conclusions

This study explored the behavior of locally available textile fabrics for civil structural applications. Sixteen types of fabrics were procured from the local market. Various tests including microscopic analysis, mass per unit area, ends and picks of fabrics, yarn number, and tensile strength of fabrics were performed in order to examine their material characteristics. Moreover, textile-reinforced mortar (TRM) specimens incorporating mortar mixture and textile fabrics were also casted and tested for determining their tensile behavior.

Microscopic analysis revealed that the tested fabrics were plain, twill (2/1) and twill (3/1) fabric weave types. Tested fabrics were composed of cotton, polyester, nylon, glass

and jute. The greater the number of yarns, the denser will be the fabric, and ultimately it will have more mass per unit area. It was observed that the tested textile fabrics have mass per unit area in the range of 117 to 1145 g/m$^2$ due to their various composition, weave type and thickness. Tested fabric TF13 had 204 yarns/inch in warp direction, which is the highest among all the tested textile fabrics. The greater the number of yarns, the greater will be the tensile strength, but thickness of yarns and fiber composition have an equivalent role.

For all the tested textile fabric specimens under uniaxial tension, the peak tensile load in warp direction was higher compared to that of the weft direction. For instance, tested specimen TF3 has 2531 and 1999 N peak load carrying capacity in warp and weft directions, respectively. This was attributed to the higher number of yarns in warp direction. Tested TF1 specimens exhibited tensile loads of 3178 and 2401 in warp and weft directions, respectively. It should also be noted that the tested plain fabric exhibited higher tensile load compared to that of the twill (2/1) and twill (3/1) fabric weave types.

The TRM test concluded that the tested specimen TRM12 had the highest tensile breaking load among all the tested specimens. It can be argued that the fabric specimens with more warp and weft spacing have stronger matrix–fabric bond, leading to improved strength properties. Lower weft spacing in the weft causes premature cracking at lower load level. In denser fabrics, less penetration of mortar will take place within the yarns, leading to poor fabric–matrix bond, and ultimately the lower the strength will be. In short, this study explored the potential and characterization of locally available textile fabrics and it will facilitate for construction stakeholders their viable application in various construction needs.

**Author Contributions:** Conceptualization, S.A. (Safeer Abbas); methodology, S.A. (Safeer Abbas), S.A. (Shahid Ali), R.A. and M.L.N.; validation, S.A. (Safeer Abbas) and M.L.N.; formal analysis, R.A. and S.A. (Shahid Ali); investigation, S.A. (Safeer Abbas) and M.L.N.; writing—original draft preparation, S.A. (Safeer Abbas) and R.A; writing—review and editing, M.L.N. and S.A. (Safeer Abbas); supervision, S.A. (Safeer Abbas) and S.A. (Shahid Ali). All authors have read and agreed to the published version of the manuscript.

**Funding:** This research received no external funding.

**Data Availability Statement:** Not applicable.

**Conflicts of Interest:** The authors declare no conflict of interest.

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
