# Peer review of "Material Characterization of Locally Available Textile Fabrics for Structural Applications"

_buildings, doi:10.3390/buildings12101589_

Round 1

Reviewer 1 Report

ID Manuscript  Number:  buildings-1924557

Title:  Material Characterization of Textile Fabrics for Structural Applications: An Economical and Sustainable Solution.

Suggested comments and corrections for the Authors of the paper are following:

The title:

The second part of the title “An economical and substantial solutions” refers not precise to the content of the article. Abstract contains only in lines 16-18 short statement about it. Similarly it is mentioned about it in the end of the Introduction does not contain the explanation about  meaning of economical and sustainable solutions. No comment in the discussion of results and no conclusion about it. It should be completed.

Section 2.

-- Line 122:  There is short information about sixteen textile fabrics procured from local market. More information could be added about criteria of selectied textile fabrics for testing, what properties were taken into consideration,  is it  fiber material, weave type, yarn number or other properties.

-- Line 125:  Fraction of fine sand could be confirmed, similarly as others components of TRM specimens.

--- Line 128:  Information could be added about chemical type of water reducer – superplasticizer.

--- Line 133  Improper name of mixing device “electric mortar mixture” is used. Maybe better  electric mortar stirrer, or agitator”.

--- Line 140:  Little information about average mortar strength. Probably it is compressive strength. How many samples, what sizes, what time, temperature and humidity of hardening were used ?

Section 3.

-- Figure 1: Because figures (b) and (c) look different, better to describe them separately and more precisely with giving symbols of different specimens according to Table 4.

--  Figure 2:  Symbol of shown textile fabric could be given,

-- Figure 3:  Better to sign (a) and (b) figures and give symbols of specimens,

--  Figure 4:  Better to sign (a), (b), (c) figures and give specimen symbols for (a, b) and (c).

--  Line 186:  According to Table 1 mortar mixtures were used, not mortar pastes. Information could be given that 16 types of TRM specimens were prepared for each type of textile fabric TF sample.

-- Because the “warp and weft directions” are important in the study, better to give any graphical signs (e.g. arrows) at image of Fig.2 or Fig.3 (or Fig.7 or in Tab.3) so that better to show this idea, to show which direction is warp and which direction is weft.

Section 4.1

-- Table 3:  What magnification of images is ? Is it the same magnification or different. To give information about magnification.

-- Because the magnification of fabric views is not large, it is not SEM analysis, the name “microstructural analysis” is not proper. Presented images of fabric surfaces are rather microscopic analysis of textile fabric textures. 

The term “microstructural analysis” used in the text should be replaced into “microscopic analysis”.

-- Looking at images different sizes of yarns can be seen, e.g. glass fabric is made of wide strips and others are made of fine strings or cords. There is no information about real sizes in cross sections of strips, cords (width, thickness, diameter).

-- Table 4:  Additional information could be given (in additional column)  to the samples of the same wave and material (TF1 – TF4), (TF7, TF8).

Section 4.2

-- Line 272:  Abbreviations HTPET and PP are used here for the first time and better full names should be given before them.

Section 4.3

-- Figure 9:  Not precise title of vertical axis, better “Number of Yarns / inch”,

-- Cited results of Ferdous and Banerjee should be completed about type of tested fabric materials.

Section 4.5

--  Lines 317, 318: Cited results of Ferdous and Banerjee could be completed about type of fabric materials.

Similarly information could be given at citation of Ferdous (Line 327) and Banerjee (Line 331).

-- Figure at page 14: Improper number, Figure 14 should be. There are no titles referring to Figures (a), (b), (c). Titles should inform about symbols of tested samples.   

 -- Line 325:  Values of maximum  deflection for TF8 and TF7 specimens in warp direction 26,25 mm and 44,95 mm are different to the values in Table 5:  28,27 mm and 43,59 mm.

-- Table 5: Results from the this Table are only shortly mentioned in the text (Line 308, 309) and partially commented for samples TF7 and TF8 according to maximum deflection. No more discussion is about the results from Table 5.

Are the peak loads the same as in Fig.11, 12, 13 ? There are differences for some samples, e.g. for TF7, TF8, TF9 weft.

Section 4.6

-- The tensile force for TRM12 is 4.18 kN and for TRM10 is 0,47 kN but it is written in line 353 that strength is 88% more of TRM12 than TRM10 specimen.

If tensile strength values are considered, additional Figure of strength values for tested TRM samples could be shown. 

Discussion is only for samples with the maximum and minimum load values. There are no comments about others TRM samples.

There are no comments about comparison of TRM tensile results to the tensile results of textile fabric specimens. Such comparison could be interesting, what differences are, is beneficial influence of mortar ?

The bar figure could be presented to show tensile load both for TF samples from Table 5 and for TRM samples.

Conclusions

-- Lines 369 – 371:  Comment about sample conditioning is not a conclusion, could be omitted,

-- There are no conclusion referring to economical and sustainable properties of tested textile fabrics. Do realized study confirm economical and sustainable properties of tested solutions ?

English:

There are some doubtful phrases to check:

-- Line 39:  “compression strength” or “compressive strength”,

-- Table 3:  “microstructural” or “microstructure”

-- Table 4:  “micro-structural” or “microstructural

Author Response

The authors would like to thank the respected reviewer for the thoughtful and insightful comments, which greatly improved the manuscript. We have attached a point-by-point explanation of how each comment was carefully addressed. We hope the revised manuscript meets the publication standards of the journal.

Reviewer 2 Report

Dear Authors,
thank you for your interesting paper focused on the structural application of textile fabrics. My comments are:
- line 145, there is given "were conditioned" - I think that it is better to use "were placed in the environment",
- chapter 2 - missing information about what the textile samples were made of - cotton? flax? silicone? glass? carbon? Were they all the same? Or different types of textiles? How many samples were made of each type of textile? Some information is given only later in chapter 4. Perhaps it would be appropriate to give Tab. 4 to chapter 2,
- line 173, the dimensions of the samples 500x50mm are stated there - how was the thickness of the samples needed to determine the tensile strength measured/determined?
- chapter. 4.5, Figs. 11-13 - is it necessary to divide these results up to three figs? Thus, not all 16 samples are displayed, but only 6 samples. How did the other samples behave?
- chapter. 4.5, Figs. 11-13 - in the text before, it was stated that only the textile samples (Fig. 4) and the textile samples in the mortar (fig. 6) were tested - the results shown in fig. 11-13 belong to which samples?
- chapter 4.6 - were samples taken from the mortar itself for comparison? Thus the contribution of textiles would be obtained/verified.
- l. 363 - it is Figure 16, not again figure 15, renumber it.

Best regards.

Author Response

The authors would like to thank the respected Reviewer for the thoughtful and insightful comments, which greatly improved the manuscript. We have attached a point-by-point explanation of how each comment was carefully addressed. We hope the revised manuscript meets the publication standards of the journal.

Reviewer 3 Report

Material Characterization of Textile Fabrics for Structural Applications: An Economical and Sustainable Solution

The article is interesting and well-written. A few minor suggestions are given below

Keywords should be improve 

Some more latest studies are required in the introduction section to further highlight the importance of this study. 

The introduction is too long.

figure 7b is not clear 

Result graphs should be improved 

Authors must summarize results in a more systematic way with reference to the previous studies.

22 to 25 references have some deviation from other reference formats. (highlighted year) 

Author Response

(The authors gave the same response as above.)

Round 2

Reviewer 2 Report

Dear Authors,

thank you for improving the paper. Now, I have no other comments.

Best regards.